# Determinants of in-hospital mortality in COVID-19; a prospective cohort study from Pakistan

Samreen Sarfaraz[1]*, Quratulain Shaikh[2]*, Syed Ghazanfar Saleem[3], Anum Rahim[4], Fivzia Farooq Herekar[5], Samina Junejo[6], Aneela Hussain[7]

1 Infectious Disease Department, The Indus Hospital, Karachi, Pakistan, 2 Indus Hospital Research Centre, The Indus Hospital, Karachi, Pakistan, 3 Emergency Department, The Indus Hospital, Karachi, Pakistan, 4 Indus Hospital Research Centre, The Indus Hospital, Karachi, Pakistan, 5 Internal Medicine Department, The Indus Hospital, Karachi, Pakistan, 6 Paediatric infectious Diseases, The Indus Hospital, Karachi, Pakistan, 7 Infectious Disease Department, The Indus Hospital, Karachi, Pakistan

* docquratshaikh@gmail.com (QS); samreen2002@gmail.com (SS)

Data Availability Statement: Data file is available from the Mendeley database (DOI:10.17632/j6jz4vk6r6.2).

## Abstract

A prospective cohort study was conducted at the Indus Hospital Karachi, Pakistan between March and June 2020 to estimate the in-hospital mortality among hospitalized COVID-19 patients and its determinants. A total of 170 adult patients were enrolled and all-cause mortality was found to be 39% (67/170). Most non-survivors were above 60 years of age (64%) while gender distribution was quite similar in both groups (males: 77% vs 78%). Most (80.6%) non-survivors came with peripheral oxygen saturation less than 93% while 95% of them had critical disease on arrival. Use of non-invasive ventilation in emergency room was higher among non-survivors (56.7%) versus survivors (26.2%). Median Interleukin-6 levels were higher among non-survivors (78.6: IQR = 33.8–49.0) compared to survivors (21.8: IQR = 12.6–36.3). Most patients in the non-survivor group (86.6%) required invasive ventilator support during hospital stay compared to 7.8% in the survivors. The median duration of ICU stay was longer for non-survivors (9: IQR = 6–12) compared to survivors (5: IQR = 3–7) days. Univariable binary logistic regression showed that age above 60 years, oxygen saturation below 93%, Neutrophil to lymphocyte ratio above 5, procalcitonin above 2ng/ml, unit increase in SOFA score and arterial lactate levels were associated with mortality. We also found that a unit decrease in Pao2/FiO2 ratio and serum albumin were associated with mortality in our patients. Multivariable regression showed that age above 60 years (aOR = 3.4: 95% CI = 1.6–6.9), peripheral oxygen saturation below 93% (aOR = 3.5:95% CI = 1.6–7.7) and serum pro-calcitonin above 2ng/ml (aOR = 4.8; 95% CI = 1.9–12.2) were associated with higher odds of mortality when adjusted by month of admission. Most common cause of death was multisystem organ failure in 35 (56.6%) non-survivors while 22 (35.5%) died due to respiratory failure. Larger prospective studies are needed to further strengthen these findings.

**Funding:** The author(s) received no specific funding for this work.

**Competing interests:** The authors have declared that no competing interests exist.

## Introduction

In December 2019 a highly transmissible respiratory illness caused by the severe acute respiratory syndrome coronavirus 2 (SARS-Cov-2) originated in Wuhan, China and caused a pandemic by its rapid spread [1]. It has resulted in 128,229,141 infections and 2,803,975 deaths globally as of 30th March 2021. Pakistan ranks 31st among the list of high burden countries with 659,116 confirmed infections and 14,256 deaths which is much lower compared to its immediate neighbours [2]. The Pakistan mortality rate of 2% [3] is comparable to that of India (1.45%) but lower than that of Iran (4.68%) and several European countries including UK (3.43%) and Italy (3.52%) [2]. The reasons for this difference in fatality is largely unknown but a multifactorial combination of viral immunogenicity, genetic makeup of the host, demography and seasonal variation may play a role in this [4]. Sind province recorded the country's first case on 26th February 2020 and since then has received 44% of the country's COVID-19 confirmed cases with the largest city Karachi being worst hit [3]. The peak of infection in the first wave was reached in mid-June when on average 7000 new infections were recorded in a day and the maximum number of deaths recorded were 153 on 20th June 2020. Major hospitals in all big cities were overwhelmed straining the health infra-structure. The Indus Hospital, Karachi emerged as a front liner with a dedicated isolation unit providing free of cost treatment to the sick COVID-19 patients requiring hospitalization. The hospital worked with the government of Sind as a diagnostic and referral center. The experience was challenging for our physicians and allied health staff. In this paper we aim to estimate the in-hospital mortality of COVID-19 in our hospital and study its determinants. To this date there is insubstantial published data on in-hospital mortality from Pakistan. As Pakistan has already entered the second wave now, these data will help in risk stratification and management of COVID-19 patients.

## Methods

This prospective cohort study was conducted at the Indus Hospital's COVID isolation unit. The Indus Hospital (TIH) Karachi is a 300 bedded tertiary care hospital set up with public private partnership which provides free of cost services to the people. The COVID unit was initially a 20 bedded dedicated COVID facility established in March 2020, later expanded to 56 beds. All COVID-19 (Nasopharyngeal PCR positive) patients aged 18 years and above admitted between 19th March and 7th June 2020 were included. Patients were enrolled into the study as soon as they were admitted in the COVID unit and were followed till death or discharge from hospital.

Demographic information, clinical presentation, laboratory abnormalities including inflammatory markers and imaging results were recorded. The primary outcome was all-cause in-hospital mortality while we also compared length of stay, occurrence of in-hospital complications, use of inotropic support and of mechanical ventilation among survivors and non survivors.

Patients were categorized as per WHO definitions into asymptomatic (COVID-19 PCR positive but with no clinical manifestation attributable to COVID-19), mild (symptomatic without evidence of pneumonia), moderate (with clinical signs of pneumonia and oxygen saturation $\geq$ 90% on room air), severe (signs of pneumonia with respiratory rate $\geq$ 30 breaths/min or SpO2 < 90% on room air) and critical (development of Acute Respiratory Distress Syndrome, septic shock or multi organ dysfunction) [5]. The study was approved by the institutional IRB under IRD_IRB_2020_04_002.

Data were recorded on a REDCap electronic data capture tool hosted at (The Indus Hospital) [6] and analyzed on Stata 14 [7]. Continuous variables were summarized by mean (SD) or median (IQR) as appropriate. Distribution of categorical variables were expressed as

percentages of the various categories (%). Univariable binary logistic regression was used to determine association of predictors with mortality. Purposeful selection method [8] was applied for building the multivariable logistic regression model and likelihood ratio tests were used to select the final model. Interaction was not tested in the final model. Multicollinearity was assessed between predictors in the final model. Goodness of fit was tested using classification table and area under curve. Crude and adjusted Odds ratios and 95% confidence intervals were reported.

## Results

The emergency department of Indus hospital, Karachi received 11,855 suspected COVID-19 patients out of which 3,851 tested positive for COVID-19 PCR (positivity rate 32.5%) from 19th March to 7th June 2020 (Fig 1). The median age of those who tested positive for COVID-19 PCR was 35 (IQR = 26–50) years. The teleconsultation service set up for COVID-19 at TIH, managed most (3,422; 88.8%) of these asymptomatic to mild spectrum patients at home through a robust algorithm based system of symptomatic treatment, follow up and counselling. The rest of the (n = 384/3851: 9.9%) patients needed admission but only 193 (50%) got admitted due to inavailability of beds on arrival, others were referred to other isolation units in the city. Seven patients who were admitted at TIH were less than 18 years old hence removed from this analysis. Some (n = 16) patients were either asymptomatic or had mild disease and needed inpatient care due to other indications like hemodialysis or emergency surgical intervention (obstructed hernia, acute appendicitis). Eventually, 170 participants were included in the cohort. The all-cause in-hospital mortality was 39% (67/170).

### Clinical presentation

Those who died had a mean age of 61 (± 12.57) years compared to those who survived 53 (± 13) years (not shown in the table). A similar gender distribution was observed in both groups Table 1 (males = 77.6% in non-survivors compared to 78.6%). Median oxygen

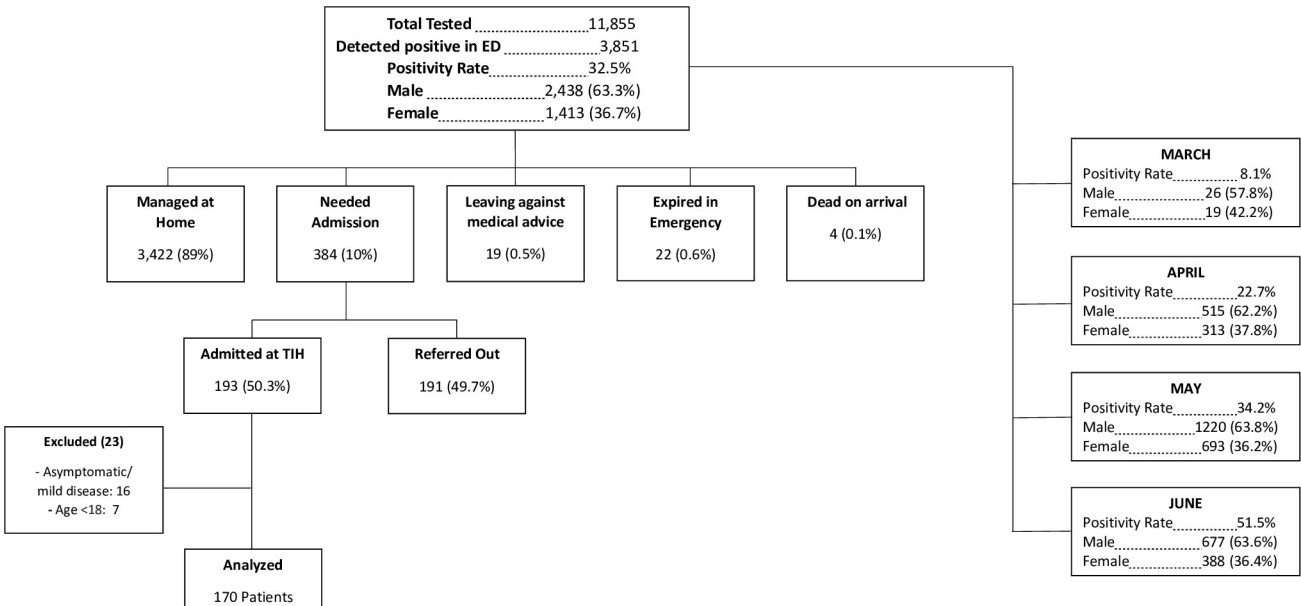

**Fig 1. Flow and outcomes of COVID-19 suspects at the Indus Hospital Emergency (19[th] March-7[th] June 2020).**

saturation in peripheral blood measured through a pulse oximeter was 86% (IQR = 71.7%-90.2%) among non-survivors versus 91.5% (IQR = 85.0% -95.2%) in survivors (not shown in the table). Most patients among non-survivors had critical disease 95.5% versus 70.9% among survivors. Use of non-invasive ventilation (NIV) in emergency room was higher among non-survivors (56.7%) versus survivors (26.2%). The Sequential Organ Failure Assessment (SOFA) [9] score was higher among non-survivors (median = 6: IQR = 5–8.2) compared to survivors (median = 4: IQR = 4–6).

Laboratory parameters are described in detail in Table 2. Neutrophil to lymphocyte ratio (NLR) was at least five in 86.6% of non-survivors compared to 70.9% among survivors. Median PaO2/Fio2 ratio was lower among non-survivors (156.8: IQR = 77–233.3) than survivors (215.5: IQR = 141.8–304.7). Inflammatory markers including C-Reactive protein (CRP), d-dimer, lactate dehydrogenase (LDH), Ferritin, Interleukin-6 (IL-6) were all higher among non-survivors compared to survivors. Serum pro-calcitonin levels were higher among non-survivors compared to survivors.

### Hospital course and outcome

Details of experimental therapies given to patients are depicted in the Table 3. Most patients in the non-survivor group required invasive ventilator support during hospital stay (86.6% vs 7.8%) while median duration of ICU stay was longer (9: IQR = 6–12) compared to survivors (5: IQR = 3–7) days. Most common cause of death was multisystem organ failure in 35 (56.6%) non-survivors while 22 (35.5%) died due to respiratory failure.

### Predictors of in-hospital all-cause mortality

Univariable binary logistic regression (Table 4) showed that age greater than 60 years, peripheral oxygen saturation less than 93%, use of NIV in emergency room, serum procalcitonin level of higher than 2 ng/ml, NLR of at least 5, high clinical severity score (SOFA, CURB 65, MuLBSTA) and high arterial lactate levels were associated with higher odds of mortality. Lower Pao2/Fio2 ratio and low serum albumin levels were associated with higher odds of mortality in our data.

Multivariable binary logistic regression (Table 5) showed that those who were older than 60 years had 3.5 times (95% CI = 1.6–7.8) the odds of mortality compared to those who were less than or equal to 60 years when adjusted for other variables in the model including month of admission. Patients who had peripheral oxygen saturation measured through pulse oximeter less than 93% had 3.5 (95% CI = 1.6–7.8) times higher odds of mortality compared to those with oxygen saturation of at least 93% in the presence of other variables in the model. Serum procalcitonin levels of more than 2 ng/ml were associated with 4.8 (95% CI = 1.9–12.4) times the odds of mortality after adjustment for other variables in the model. Final model was adjusted for month of admission. Receiver operating curve (ROC) analysis revealed that the model has a sensitivity of 60.9% and specificity of 80.2% in predicting mortality among hospitalized COVID-19 patients and the correct classification rate was 72%. The Area under the curve (AUC) of the model was 0.76.

### Discussion

In this prospective cohort study, we report the clinical attributes and risk factors associated with all-cause mortality among hospitalized COVID-19 patients. We found the all-cause mortality to be 39% which is disproportionately high in those who were ventilated (58/66: 88%). It is important to highlight here that our most of our patients suffered from critical disease (approximately 80.5% of total admitted) with 38% needing non-invasive ventilation (NIV) in

**Table 1. Clinical characteristics of study participants by mortality.**

|  | Non-Survivors | Survivors |
|---|---|---|
|  | n(%) | n(%) |
| **Age (years)** |  |  |
| ≤60 | 24 (35.8) | 64 (62.1) |
| >60 | 43 (64.2) | 39 (37.9) |
| **Gender** |  |  |
| Male | 52 (77.6) | 81 (78.6) |
| Female | 15 (22.4) | 22 (21.4) |
| **Symptoms** |  |  |
| Fever | 54 (80.6) | 86 (63.5) |
| Cough | 40 (59.7) | 62 (60.2) |
| SOB | 57 (85.1) | 83 (80.6) |
| Runny nose | 1 (1.5) | 3 (2.9) |
| Sore Throat | 3 (4.5) | 4 (3.9) |
| Chest Pain | 5 (7.5) | 2 (1.9) |
| Fatigue | 8 (11.9) | 5 (4.9) |
| Diarrhea | 4 (6.0) | 8 (7.8) |
| Vomiting | 2 (3.0) | 10 (9.7) |
| Others | 13 (19.4) | 26 (25.2) |
| **Duration of symptoms* (days)** | 6 (4–7) | 7 (4–10) |
| **Comorbid conditions** |  |  |
| None | 13 (19.4) | 38 (36.9) |
| Hypertension | 35 (52.2) | 39 (37.9) |
| Diabetes | 35 (52.2) | 41 (39.8) |
| Liver Disease | 1 (1.5) | 0 (0) |
| Lung Disease | 3 (4.5) | 3 (2.9) |
| Renal Disease | 6 (9.0) | 7 (6.8) |
| Heart Disease | 7 (10.4) | 10 (9.7) |
| Other | 12 (17.9) | 26 (25.2) |
| **Number of comorbid conditions** |  |  |
| < = 2 | 43 (79.6) | 45 (69.2) |
| 3 or more | 11 (20.4) | 20 (30.8) |
| **Clinical Signs at presentation** |  |  |
| **Systolic Blood pressure* (mm/Hg)** | 144 (123–159) | 135 (122–149) |
| **Diastolic Blood pressure* (mm/Hg)** | 80 (68.5–95) | 79 (72.5–89) |
| **Pulse**/minute** | 104.0 ± 22.0 | 101.1 ±20.3 |
| **Respiratory Rate*/m** | 32 (26–38) | 28 (23–32) |
| **Temperature* °F** | 98.6 (98–99.0) | 98.4 (98–98.6) |
| **Oxygen Saturation (%)** |  |  |
| ≥93% | 13 (19.4) | 46 (44.7) |
| <93% | 54 (80.6) | 57 (51.4) |
| **Glasgow Coma Scale** | 15 (15–15) | 15 (15–15) |
| **Disease Severity** |  |  |
| Moderate | 0 (0) | 18 (17.5) |
| Severe | 3 (4.5) | 12 (11.7) |
| Critical | 64 (95.5) | 73 (70.9) |
| **Non-invasive ventilation in emergency room** |  |  |
| Yes | 38 (56.7) | 27 (26.2) |

(*Continued*)

**Table 1.** (Continued)

|  | Non-Survivors | Survivors |
|---|---|---|
|  | n(%) | n(%) |
| **Age (years)** |  |  |
| No | 29 (43.3) | 76 (73.8) |
| **Clinical Severity Scores** |  |  |
| SOFA score* | 6 (5–8.2) | 4 (4–6) |
| CURB-65 score* | 2 (1–3) | 1 (0–2) |
| MuLBSTA* | 7.5 (6–9.5) | 5 (2–8) |
| Month of admission |  |  |
| March | 4 (5.9) | 2 (1.9) |
| April | 22 (32.8) | 28 (27.4) |
| May | 32 (47.8) | 46 (45.1) |
| June | 9 (13.4) | 26 (25.5) |

*Median (Q1-Q3)

**Mean±SD.

emergency room to manage respiratory failure. ICU admission was needed for 59% (95/160). Approximately, 38.8% (66/170) required mechanical ventilation (MV) hence our patient population appears to be sicker compared to the only other unpublished data from the city [10]. We believe, being a private referral center they may have admitted milder spectrum patients for the purpose of isolation and monitoring. On the contrary, as mentioned before, our center is a philanthropic, free of cost referral centre for the underprivileged with limited bed capacity. Hence, admission was strictly reserved for sick patients requiring in-hospital management. The poor survival in ventilated cases, apart from the natural disease process, may be due to little knowledge of the pathogenic mechanism of respiratory injury and its management in the initial days. Most patients were managed with early invasive ventilator support to avoid fatigue and potential risk of aerosolization of COVID-19 with NIV [11]. Gradual understanding of the disease process has shifted the management strategy from early mechanical ventilation towards use of NIV till tolerated as suggested by the National and International COVID guidelines [3, 5]. Global mortality from COVID-19 varies widely (20% -97%) [12, 13] depending on ICU facilities, ventilator performance, experience of ICU team, patient and disease characteristics, geographic area, seasonality and duration of follow up [4]. High ventilator mortality has been reported even from the best centers in Wuhan (97%), New York (88%), UK (67%) and Italy (53.4%) [14–17] questioning the role of invasive ventilation in COVID-19 management especially in lower middle income countries [18]. Ventilator induced lung injury due to barotrauma, volutrauma, atelectrauma, oxytrauma and infections further jeopardize the outcome of COVID-19 patients [19]. Hospitalized patients with COVID-19 have 5 times higher reported mortality than those with influenza pneumonia [20].

Non-survivors in our study showed worse clinical profile with low peripheral oxygen saturation, respiratory rate and raised inflammatory parameters at presentation. This indicates that they were already in the late phases of Cytokine Release Syndrome (CRS) at admission [21]. The median time to hospitalization from onset of symptoms was similar for both survivors and non-survivors (6 days vs 7 days) consistent with reported literature [22]. However, why some patients were more prone to develop CRS by day 7 is not clear. It is important to note that the overall COVID-19 cohort presenting to our hospital (3851 patients) was nearly 20 years younger (median age 35 vs 58 years) than the subgroup who got admitted [23]. Older

**Table 2. Baseline laboratory parameters by survival status.**

| | Non-Survivors Median (IQR) | Survivors Median (IQR) |
|---|---|---|
| Hemoglobin(gm/dl) | 12.7 (11.1–14.1) | 13.10 (11.6–14.1) |
| WBC Count (x10E9/L) | 12.0 (8.3–15.1) | 9.8 (7.2–13.9) |
| Platelet Count (x10E9/L) | 198.0 (147.0–296.0) | 239.0 (190.0–331.0) |
| Absolute neutrophil count (ANC) (x10E9/L) | 10.3 (7.6–13.9) | 7.9 (5.5–11.5) |
| Absolute Lymphocyte count (ALC) (x10E9/L) | 1.32 (0.53–1.32) | 1.07 (0.74–1.69) |
| Neutrophil to Lymphocyte ratio (NLR)* | | |
| <5 | 9 (13.4) | 30 (29.1) |
| ≥5 | 58 (86.6) | 73 (70.9) |
| Arterial pH | 7.4 (7.3–7.4) | 7.4 (7.4–7.4) |
| PCO2 (mmHg) | 31.3 (27.5–35.8) | 31.9 (29.0–34.2) |
| PO2 (mmHg) | 50.2 (41.3–64.7) | 64.3 (53.8–77.8) |
| PaO2/FiO2 Ratio | 156.8 (77–233.3) | 215.5 (141.8–304.7) |
| Urea (mg/dl) | 44.0 (32.0–67.5) | 34.0 (24.0–51.0) |
| Creatinine (mg/dl) | 1.2 (0.9–1.8) | 1.0 (0.8–1.3) |
| Sodium (mg/dl) | 136.0 (133.0–139.0) | 137 (133.0–139) |
| Potassium(mg/dl) | 4.0 (3.6–4.6) | 4.2 (3.9–4.6) |
| Bicarbonate (mg/dl) | 19.5 (16.0–21.7) | 20 (18–22) |
| Total Bilirubin (mg/dl) | 0.6 (0.4–0.9) | 0.6 (0.4–0.8) |
| SGPT (U/L) | 43.5 (22.0–90.2) | 31.0 (15.5–52.0) |
| Arterial Lactate (mmol/L) | 2.3 (1.8–3.3) | 1.3 (1.1–2.0) |
| Serum Albumin (g/L) | 3.2 (2.6–3.4) | 3.5 (3.1–3.7) |
| Prothrombin Time (sec) | 11.2 (10.4–12.8) | 11.2 (10.7–11.7) |
| APTT (sec) | 29.8 (26.8–32.4) | 29.9 (23.9–33.7) |
| C-Reactive Protein mg/L | 192.2 (86.9–337.0) | 111.3 (60.7–220.3) |
| D-dimer ng/ml | 1860.5 (728.7–5767.5) | 1277.0 (622.0–5688.0) |
| Lactate Dehydrogenase (LDH) U/L | 549.5 (469–855) | 476 (355–694) |
| Ferritin ng/ml | 1315 (532.7–1675.5) | 1227.0 (292.7–1675.5) |
| Interleukin-6 (IL-6) pg/ml | 78.6 (33.8–490.0) | 21.8 (12.6–36.3) |
| Procalcitonin ng/ml | 0.6 (0.2–2.2) | 0.2 (0.1–0.6) |
| Troponin ng/ml | 16 (6.7–53.5) | 7 (4–18) |
| Blood culture* | | |
| Positive | 5 (8.3) | 7 (8.3) |
| Negative | 55 (91.7) | 77 (91.7) |
| Chest x-ray* | | |
| Unilateral Radiologic Findings | 3 (4.5) | 4 (3.9) |
| Bilateral Radiologic Findings | 59 (88.1) | 85 (82.5) |
| Multilobar Infiltrates | 36 (53.7) | 28 (27.2) |
| Consolidation | 32 (47.8) | 32 (31.1) |
| Pleural Effusion | 3 (4.5) | 4 (3.9) |
| Others | 7 (10.4) | 7 (6.8) |
| Normal | 0 (0) | 10 (9.7) |

*Frequency (%).

age was also found to be associated with higher mortality in our data as reported globally [24, 25]. Age above 60 years has consistently shown to be relevant to mortality since the beginning of the pandemic and our results were consistent in identifying this as a predictor of mortality among others.

**Table 3. Hospital course by mortality.**

| | Non-Survivors n (%) | Survivors n (%) |
|---|---|---|
| **Treatment** | | |
| **Methyl Prednisolone** | | |
| Yes | 55 (82.1) | 82 (82.0) |
| No | 12 (17.9) | 18 (18.0) |
| **Antibiotics** | | |
| Yes | 63 (94.0) | 82 (81.2) |
| No | 4 (6.0) | 19 (18.8) |
| **Anticoagulation** | | |
| Therapeutic doses | 43 (64.2) | 35 (35.0) |
| Prophylactic doses | 18 (26.9) | 57 (76.0) |
| None | 6 (9.0) | 8 (8.0) |
| **Hydroxychloroquine (HCQ)** | | |
| No | 27 (40.3) | 56 (54.4) |
| Yes | 40 (59.7) | 47 (45.6) |
| **Azithromycin (AZT)** | | |
| No | 30 (44.8) | 67 (65.0) |
| Yes | 37 (55.2) | 36 (35.0) |
| **Tocilizumab (TCZ)** | | |
| Single Dose | 13 (19.4) | 23 (22.3) |
| Two Doses | 8 (11.9) | 1 (1.0) |
| Not Given | 46 (68.7) | 79 (76.7) |
| **IVIG** | | |
| Single Dose | 14 (20.9) | 7 (6.8) |
| Multiple Doses | 5 (7.5) | 0 (0) |
| Not Given | 48 (71.6) | 96 (93.2) |
| **Admission to ICU** | | |
| Yes | 61 (93.8) | 34 (35.8) |
| No | 4 (6.2) | 61 (64.2) |
| **Invasive Ventilation** | | |
| Yes | 58 (86.6) | 8 (7.8) |
| No | 9 (13.4) | 95 (92.2) |
| **In-Hospital complications** | | |
| None | 10 (14.9) | 71 (68.9) |
| Cardiac | 17 (25.4) | 10 (9.7) |
| Nosocomial Infection | 20 (29.9) | 11 (10.7) |
| CNS | 3 (4.5) | 2 (1.9) |
| Septic Shock | 30 (44.8) | 2 (1.9) |
| MODS | 23 (34.3) | 0 (0) |
| AKI | 39 (58.2) | 7 (6.8) |
| Thromboembolism | 6 (9.0) | 1 (1.0) |
| Barotrauma | 3 (4.5) | 1 (1.0) |
| DIC | 9 (13.4) | 0 (0) |
| Sever Hyperglycemia | 4 (6.0) | 3 (2.9) |
| Electrolyte Imbalance | 11 (16.4) | 4 (3.9) |
| Other | 7 (10.4) | 7 (6.8) |
| **Length of Hospital stay* (days)** | 9 (6–12) | 13 (11–15) |
| **-Length of ICU Stay* (days)** | 9 (6–12) | 5 (3–7) |

(*Continued*)

**Table 3.** (Continued)

| | Non-Survivors n (%) | Survivors n (%) |
|---|---|---|
| **Days of Intubation**[*] | 6 (3–9) | 4.5 (2–7) |

[*]Median (IQR).

Neutrophilia and thrombocytopenia were more pronounced among non-survivors with NLR of at least 5 was seen more commonly among non-survivors than survivors (Table 2). Wu C et al found a significant association between neutrophilia, lymphopenia (peripheral CD3, CD4, and CD8 T-cell counts decreased) and development of Acute Respiratory Distress Syndrome (ARDS) [26]. As observed in literature, CRP, D-dimer, LDH, Ferritin and IL-6 were all higher among non-survivors in our cohort [26]. Non-survivors in our cohort developed acute kidney injury, sepsis and multi organ dysfunction syndrome (MODS) more frequently than survivors. Unit increase in SOFA score and CURB 65 was associated with higher odds of mortality in our data. Since SOFA tends to reflect the effect on multiple organ systems, it has proved to be a better predictor of mortality in COVID-19 in previous literature [27]. These were not retained in the final model.

Use of experimental therapies including steroids, antibiotics, anticoagulation, Hydroxychloroquine (HCQ) during the first wave was similar in both survivors and non-survivors.

**Table 4. Univariable binary logistic regression for predictors of in-hospital all-cause mortality.**

| | Crude OR | 95% CI | p-value |
|---|---|---|---|
| Age | | | |
| ≤60 years | 1 | | |
| >60 years | 2.9 | 1.5–5.6 | 0.001 |
| Gender | | | |
| Female | 1 | | |
| Male | 0.9 | 0.4–2.0 | 0.874 |
| Oxygen saturation | | | |
| ≥93% | 1 | | |
| <93% | 3.3 | 1.6–6.9 | 0.005 |
| SOFA | 1.4 | 1.2–1.7 | 0.001 |
| CURB-65 | 1.8 | 1.3–2.5 | 0.001 |
| MuLBSTA | 1.2 | 1.1–1.4 | 0.000 |
| (NLR) | | | |
| <5 | 1 | | |
| ≥5 | 2.6 | 1.2–6.0 | 0.020 |
| Pao2/Fio2 ratio | 0.5 | 0.4–0.8 | 0.000 |
| Arterial lactate (mmol/L) | 1.5 | 1.1–2.0 | 0.012 |
| Albumin g/L | 0.3 | .13- .66 | 0.003 |
| Procalcitonin | | | |
| ≤ 2 ng/ml | 1 | | |
| >2 ng/ml | 3.6 | 1.6–8.1 | 0.002 |
| NIV in Emergency room | | | |
| No | 1 | | |
| Yes | 3.6 | 1.9–7.1 | 0.000 |

NLR-neutrophil to lymphocyte ratio, NIV-Non-invasive ventilation.

**Table 5. Multivariable Logistic regression model of all-cause mortality in COVID-19 patients.**

|  | Adjusted Odds ratios* | 95% CI | p-value |
|---|---|---|---|
| Age |  |  |  |
| ≤ 60 years | 1 |  |  |
| > 60 years | 3.4 | 1.6–6.9 | 0.001 |
| Oxygen saturation |  |  |  |
| ≥ 93% | 1 |  |  |
| < 93% | 3.5 | 1.6–7.7 | 0.002 |
| Procalcitonin |  |  |  |
| ≤2 ng/ml |  |  |  |
| >2 ng/ml | 4.8 | 1.9–12.2 | 0.001 |

*Adjusted for month of admission.

Among them, Hydroxychloroquine despite inhibiting viral replication in vitro [28] did not prove beneficial in RCTs [29], rather proved to be toxic. This led to the U.S. FDA revoking its emergency use authorization in June 2020 [30]. Use of therapeutic anticoagulation, two doses of Tocilizumab and Azithromycin was more frequent among non-survivors in our data. Azithromycin is the only effective oral drug for treatment of XDR salmonella [31] and COVID-19 pandemic has led to its mass injudicious use, both over the counter and prescription driven, which may increase antimicrobial resistance in the long run. Studies have failed to demonstrate any benefit of AZT alone [32] or in combination with HCQ [33] with a risk of compounding cardiac toxicity by QTc prolongation [34]. Antibiotics were started in 87.6% of our patients suspecting respiratory bacterial co-infection although only 7% initial blood cultures were positive (most patients were not producing sputum) and markers like WBC count and procalcitonin were not elevated in the majority. Literature reports low rates of secondary infection with COVID-19 (only 8% in a review of 9 studies) with paradoxical high consumption (72%) of broad spectrum antibiotics [35]. We believe COVID-19 specific antibiotic stewardship guidance is essential to stop the rampant over- use of antibiotics especially in LMIC countries like Pakistan where antimicrobial resistance (AMR) is already high.

Steroids have shown benefit in severe and critical COVID-19 in the RECOVERY trial [36] and a recent meta-analysis of 7 trials conducted on 1703 patients showed a reduction in 28-day mortality compared with standard care or placebo (32% vs 40%, OR = 0.66, 95% CI = 0.53–0.82) [37]. However, most experience is with dexamethasone and not methyl prednisolone (MP) although MP has been recommended as an alternate to dexamethasone in a dose of 32mg/d in current guidelines [38]. Our patient population was given MP but in higher doses (40mg q8hrs) which may have resulted in the observed hyperglycemia, secondary infections and electrolyte imbalance in our data. Moreover, a randomized trial on severe COVID-19 patients in Brazil did not show any mortality benefit of MP at 28 days as compared to placebo (37% vs 38%) [39]. Some observational studies have reported mortality benefit of TCZ [40, 41] but RCTs failed to demonstrate any difference in survival when compared to placebo or usual care [42]. Data from our center also did not show any survival benefit of TCZ [43].

Among other predictors, a peripheral oxygen saturation at presentation below 93% and use of assisted ventilation in emergency room was associated with mortality in our data. These indicators of severity of the COVID pneumonia are now universally used in guidelines [3, 5]. A high NLR is predictive of disease severity as shown by Yang et al [44] and Nalbant et al [45] among many others. Our data also shows that an NLR of at least 5 is associated with 2.6 (95% CI: 1.2–6.0) times the odds of mortality. Bacterial superinfection in COVID-19 can trigger a

cascade of multiorgan failure through sepsis and further decrease the probability of recovery. Although baseline procalcitonin was low in the overall cohort (median = 0.4 ng/ml), levels above 2 ng/ml were associated with 3.6 (95% CI: 16.-8.1) times the odds of mortality in our patients. It is uncertain whether this suggests secondary bacterial infection or hyper-inflammation as studies have suggested that raised procalcitonin as a marker of bacterial infection tends to lose specificity as COVID progresses [25, 26]. Pao2/FiO2 ratio has been shown to predict mortality as it relates to the severity of lung involvement and impairment of gas exchange [46]. Our data shows a 50% reduction in mortality with unit increase in Pao2/FiO2 ratio (OR = 0.5:95% CI = 0.4–0.8). Serum albumin is a marker of nutritional status and usually used to depict chronic undernutrition. However, it has shown a strong association with mortality in COVID-19 as suggested by Rica et al [47]. Unit increase in serum albumin in our patients was associated with 70% decreased odds of mortality (OR = 0.3; 95% CI = 0.13–0.66). Factors which remain significant in the final model were age above 60 years, SpO2 less than 93% and procalcitonin above 2 ng/ml adjusted for month of admission to account for differences in management strategies and the potential change in virus genotype over the those months.

This is the first prospective cohort study from Pakistan on in-hospital mortality of COVID-19 patients. Detailed clinical history, laboratory parameters and therapeutics have been compared. There is no attrition as patients were followed till discharge or death. Despite the limitations of a small sample our data revealed some important predictors of mortality in our study population. The study is limited by data from a single-center with critically ill COVID-19 patients which may introduce a selection bias and inflate the mortality. Hence, results from this study may help in the risk stratification and management of similar critically ill patients only. We cannot say whether patients who were referred out were sicker than our cohort or vice versa and hence results otherwise would have changed. The readers should be mindful of measurement errors in laboratory parameters as single readings were taken and validation systems vary across laboratories. The sample size is limited because of the number of beds available at the unit during the first wave. We plan to add more subjects in future as we are currently seeing the second wave. A larger multi-center cohort study from various hospitals of the country would help to further validate the findings of our study.

## Acknowledgments

We would like to thank Indus Hospital Research Center for assistance in database designing and data collection.

## Author Contributions

**Conceptualization:** Samreen Sarfaraz, Syed Ghazanfar Saleem, Fivzia Farooq Herekar, Samina Junejo, Aneela Hussain.

**Data curation:** Samreen Sarfaraz, Syed Ghazanfar Saleem, Anum Rahim, Fivzia Farooq Herekar, Samina Junejo, Aneela Hussain.

**Formal analysis:** Quratulain Shaikh, Anum Rahim.

**Methodology:** Samreen Sarfaraz, Quratulain Shaikh, Anum Rahim, Samina Junejo.

**Project administration:** Samreen Sarfaraz, Quratulain Shaikh, Anum Rahim.

**Resources:** Quratulain Shaikh, Syed Ghazanfar Saleem, Aneela Hussain.

**Software:** Quratulain Shaikh.

**Supervision:** Samreen Sarfaraz, Quratulain Shaikh.

**Writing – original draft:** Samreen Sarfaraz, Quratulain Shaikh.

**Writing – review & editing:** Samreen Sarfaraz, Quratulain Shaikh, Syed Ghazanfar Saleem, Anum Rahim, Fivzia Farooq Herekar.

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
