## [Decision Letter · Decision Letter 0]

19 Feb 2021

PONE-D-21-00091

Determinants of in-hospital mortality in COVID-19; a prospective cohort study from Pakistan

PLOS ONE

Dear Dr. Quratulain Shaikh,

Thank you for submitting your manuscript to PLOS ONE. After careful consideration, we feel that it has merit but does not fully meet PLOS ONE’s publication criteria as it currently stands. Therefore, we invite you to submit a revised version of the manuscript that addresses the points raised during the review process.

ACADEMIC EDITOR: The reviewers have raised a number of points which we believe major modifications are necessary to improve the manuscript, taking into account the reviewers' remarks. Please consider and address each of the comments raised by the reviewers before resubmitting the manuscript. This letter should not be construed as implying acceptance, as a revised version will be subject to re-review.

We look forward to receiving your revised manuscript.

Kind regards,

Wisit Cheungpasitporn, MD

Academic Editor

PLOS ONE

Journal Requirements:

2) We note that you have stated that you will provide repository information for your data at acceptance. Should your manuscript be accepted for publication, we will hold it until you provide the relevant accession numbers or DOIs necessary to access your data. If you wish to make changes to your Data Availability statement, please describe these changes in your cover letter and we will update your Data Availability statement to reflect the information you provide.

Reviewers' comments:

Reviewer's Responses to Questions

**Comments to the Author**

1. Is the manuscript technically sound, and do the data support the conclusions?

Reviewer #1: No

Reviewer #2: Partly

Reviewer #3: Partly

2. Has the statistical analysis been performed appropriately and rigorously? 

Reviewer #1: No

Reviewer #2: Yes

Reviewer #3: No

3. Have the authors made all data underlying the findings in their manuscript fully available?

Reviewer #1: No

Reviewer #2: No

Reviewer #3: Yes

4. Is the manuscript presented in an intelligible fashion and written in standard English?

Reviewer #1: Yes

Reviewer #2: Yes

Reviewer #3: Yes

5. Review Comments to the Author

Reviewer #1: The paper is well writtem and as noted by the authors in the introduction, these data could help in risk stratification and management of COVID-19 patients. The main issue of the paper resides in the fact that the treatments assignments are not randomized so that any conclusion regarding the association between treatment and survival is dubious. There are also issues in the presentation of the statistical analysis that should be (easily) corrected.

I would recommend to the authors to state clearly the assumptions they want to investigate and a major revision of the statistical analysis before any resubmission.

Statistical aspects

-------------------

It is not clear whether the study followed a pre-speficied statistical analysis plan. For example, was the criteria that led to the exclusion of patients under 15 set before the start of the data analysis? If so, were the various pre-defined steps registered anywhere?

p11 l 99 "Among the rest 384 (9.9%) needed admission but due to limited bed availability in the isolation unit, only 193 (50%) were admitted while the rest were referred to other isolation units in the city."

What was the rationale underlying the choice of patients referred to other isolation units?

Also it appears later in the text and table 1 that some of the 186 patients admitted were asymptomatic (n=2) or presenting mild symptoms (n=13). Please expain.

The statistical analysis reported at lines 110-142 table 1&2 is supported by various p-values. Those p-values have to be taken with a grain of salt since we are in the classical setting of multiplicity or multiple testing. With a number of hypotheses tested large enough, there will always be some significant p values by chance alone. Therefore, it would be good to report which p-values are significant after Benjamini-Hochberg correction.

Table 3 and l 144-158 the assignment of treatment being non randomized, it is difficult to interpet any association between treatment and outcome.

Besides, table 3 report various frequencies of treatment conditionnal on survival status. Even in case of an RCT, such numbers would be difficult to interpret. What is required indeed is a frequency of survival conditionnal on treatment.

For those two reasons, a conclusion like l. 155 "There was no mortality benefit of using methylprednisolone in our patients (p>0.05)" is not grounded at all.

This table should be left as is and would stand as a purely descriptive report of the pathways of cares.

A table of frequency of survival conditional on treatment should be provided, but again, any attempt to draw conclusion about the observed association should be avoided.

l 160 Please explain better what is meant by "Univariable binary logistic regression (Table 4) at a cut-off of p<0.10, showed that ...".

Do you mean that you implemented a logistic regression for all variables and present here only the results for variables with the lowest p-values?

Table 4: this table mixes variables of different types: demographic parameters, clinical characteristics at admission, baseline lab parameters and treatments. The treatment being non-randomized, reporting and commenting a p-value for treatment variables makes little sense. Please also add a column with the significance obtained by the Bonferroni-Hochberg procedure.

l 170-175: can you please rephrase/explain what multivariate logistic regression was implemented and what was the goal/rationale for it. Again, it seems to mix variables of different types.

I understand the idea of implementing a multivariate logistic regression to predict the survival status of of a patient at admission, but such a an approach should involve not treatment variables.

What is the goal of the multivariate regression involving all the variables?

Specific comments

-----------------

p2 Funded/unfunded study statement missing

p4 Human Subject Research (involving human

participants and/or tissue) missing

p9 l52 "Our mortality rate of 2% (3) is comparable to the India (1.45%) but lower"

-> "The Pakistan mortality rate of 2% (3) is comparable to that of India (1.45%) but lower than that of Iran (4.68% )"

p9 l58 "The 58 peak of infection in the first wave reached in..."

-> The 58 peak of infection in the first wave was reached in...

p10 Do I understand correctly that asymptomatic patients or with mild symtoms were included in this study?

p10 l88 "variables were expressed as"

-> variables were summarized by

p 20 l88 "Categorical variables were expressed as number (%)."

-> "Distribution of categorical variables were expressed as percentages of the various categories."

p10 l89 "Student’s t-test or Mann Whitney U test were used to compare continuous data. Chi square or Fischer’s Exact test was used to compare categorical data."

This statement is a bit vague at this step

p10 l91 "All variables with p-value <0.10 were considered for multivariable binary logistic regression model in order of significance." I do not not understand the last bit of this sentence "in order of significance."

p11 l101 "Seven patients who were admitted at TIH were less than years old hence removed from this analysis." Please explain why.

p11 l113 "fatigue was common in non-survivors and vomiting was seen in survivors (p>0.05)"

It is not clear to which test this pvalue refers to. Please clarify.

Reviewer #2: The information included in this article is of critical and immediate importance. While the sample size may be relatively small, it is only through publication of work such as this that we can build the literature to see the bigger picture. Because of its importance, and the critical eye on science in this area, it is crucial that the results be communicated effectively and scientifically.

In that vein, I would ask the authors to take a second look at the overall manuscript and clean up small errors and inconsistencies. Most importantly, the statistical interpretation of the data needs improvement. There is an over-reliance on p-values of more or less than a threshold and not nearly enough attention paid to the data itself, which is the more important interpretation. The p-value tells us only whether what we are seeing is likely to be a random result; it cannot speak to the magnitude or importance of an effect.

Introduction

- Could use another run through by the authors to tighten up language and remove small errors; but is intelligible.

- The end of the introduction needs to be more direct about what is being estimated. From Line 64, you aren’t estimating in-hospital mortality of Covid-19 in Pakistan, but in the Indus Hospital’s COVID isolation unit.

Methods

- The Methods section needs to be re-organized into paragraphs, currently the information jumps around topically from one sentence to the next making it hard to follow. Suggest:

o Line 74, put IRB approval on its own at the end

o Line 81, a description of the types of patients admitted should immediately follow Line 73 where you describe who was included in the study

o In a new paragraph, Lines 75-87 (apart from the above) describe how the patients were recorded and followed and make a single coherent paragraph

o From Line 87 to the end (“Data was recorded…”) this should be a new paragraph on the data analysis.

Results

- Line 99 – how was it decided who needed admission? What was the criteria? I think it is moderate to severe, but this should be stated more plainly.

- A brief background on the teleconsultation service and its role, could go in introduction or methods

- Line 99 – in Figure 1 it is indicated 9.97 needed admission, this should be rounded to 10.0 not 9.9.

- Percentages should be displayed consistently – some are rounded to whole numbers and other to 00.0%. If significant digits, that would be fine, too, but that isn’t consistent either. Even if the number is, for example, 36.0% the display should be consistent.

- Lines 101-102: the 7 patients <18 & excluded from the analysis need to be included on Figure 1 with a final box for the included cohort for analysis.

- Figure 1 – what does LAMA mean?

- Table 1 (and all others) put a space between n & (%), e.g. 8 (12.1%); true also of mean (SD) and median (IQR)

- Table 1 – headers of n (%) are not applicable to continuous variables – needs to be stated in the table what 61.4 (54.1-69.1) are. The use of ^ and ^^ is non-standard and not clear enough. It appears that even the authors got mixed up (see next point).

- Table 1 / Line 103-104 – the text says that 61.4 is a median, but the table says 61.4 is a mean, using the ^^ nomenclature defined at the bottom of the table.

- Line 103-104 and 110-111 is the same sentence repeated

- Line 111 [incorrect numbers, conclusion]– the statement that males are more likely to die because there are 77.6% males in the non-survivors compared to 76.5% males in the survivors, p<0.05 is nonsensical and at odds with what is presented in Table 1 where there is a p-value of 0.859 comparing the distribution of gender in the survivors to the distribution of gender in the non-survivors.

- Lines 112-113, what does the p>0.05 refer to? Why not put the actual p-value? But more importantly, I am unsure the authors are correctly interpreting what each p-value in Table 1 represents. In the case of the Symptoms, the p-value in Table 1 using the chi-square method only says that across ALL of the symptoms presented among the survivors and non-survivors, there is not a statistically significant difference in this distribution. It does NOT relate to which are the three most common symptoms in both groups. Also, the statement about the 3 most prevalent symptoms does not need a p-value.

- Lines 114-115, again, I cannot decipher what this p-value is supposed to represent or where it came from.

- Line 118 – I find the conclusion that temperature among non-survivors was higher suspect based on the data presented and that fact that the p-value is marked by a “D” which is not defined by the authors. Also, if this finding is real, then address the fact that differences were only by a few tenths of a degree.

- Tables – there are some errant “a” and “b” in the tables which appear to have no discernable meaning.

Not continuing line by line, the authors need to consult a statistician on the interpretation of p-values. Furthermore, their utility in this kind of paper is not very high. The authors would do better not to try to repeat all of the information in the tables in the results, but only to highlight the findings of most interest. Each sentence in the results does not need a p-value. Results such as prevalence of certain symptoms are of interest on their own and do not neatly adhere to a p-value framework.

Table 4 - For binary variables, the data provided in prior tables is sufficient to re-run the univariable logistic regression models. Having done this for a few, there may be transcribing errors. I used R and not Stata, but for example, for Gender (male) the OR I get is 1.07 after appropriate rounding of 1.067, the 95% CI I get is 0.53 – 2.22 after appropriate rounding of 0.528 and 2.218. These differences could be due to different statistical programs being run. However, there are also inconsistent numbers of digits displayed for the 95% CIs and the p-values.

Figure 2 - The depiction of multivariable logistic regression coefficients (ORs) as a forest plot is not helpful. It would be better to display this information in a table. Forest plots such as the one included are typically used to evaluate effect estimates across different studies, not within a single model.

Interpretation – the authors need to re-assess their interpretation of their OR results for both the univariable and multivariable models. All values with a significant p-value are not “associated with high odds of mortality”. A large effect size along with a significant p-value can be used to establish an association with a high odds of mortality, but a statistically significant p-value alone is not enough. More attention should be paid to the meaning of the OR and its interpretation and less to p-values. NOTE: in the abstract the phrase ‘higher odds of mortality’ is used – this is much more accurate.

Discussion

The discussion appears to be more carefully written. It focuses on important findings and does not over extend the interpretation of p-values (e.g. "The final model shows that the factors associated with mortality are…" which is an accurate interpretation). It could still go further if effect size (i.e. the value of the OR matters!) are interpreted.

Can the authors address any potential bias from patients being referred to other centers? Were more severe patients retained? Or is it assured that these were random depending only on available beds at the time of entry?

Can the authors address the reliability of the data? Was there any missing data for patients included – i.e. this being real world data could tests performed have failed to be recorded? You included many continuous laboratory values, what is the likelihood of measurement error? Measurement error is likely to reduce the observed associated with mortality. Finally, is it possible for the categorical values to have any important misclassification? As it is expected this information will be used in larger meta-analyses and further research, inclusion of any such limitations may be invaluable.

Abstract

The abstract is also over-reliant on p-value thresholds. It also says that those who died were more likely to be males, implying that they were more likely to be males than those who did not die – which is not supported by the data (see above).

Reviewer #3: I appreciate the intent of this research and the need for it to be both publicly available and scrutinized in the details before publication. There are some minor typographic concerns as well as a general need to define terms and acronyms used throughout the manuscript (at least a supplement or in first use – see GCS for example). More substantively, there are two general issues and then some more specific minor issues of note related to the statistical results presented that listed below.

First, there are many p-values reported (at least 70 unique tests), (Table 1: 16, Table 2: 40, Table 3: 14, Table 4 (some redundant but different tests used, 45) with no control for inflated Type I error rates when “fishing” across this many different response/predictor variables. At least some of these are spurious detections without some sort of control like Bonferroni or False Discovery Rate adjustments (at least within a table or suite of similar sort of tests). I would suggest doing this for the limitation of chance false discoveries and because it would simplify the final reporting of results, if only those that are selected to have small p-values are discussed. Related to this is the redundancy of testing for survivor/not vs each variable as a response (Chi-square/Fisher’s are non-directional but the others are not) and then repeating the same tests in the univariate logistic regression models with survivor/not as the response. The summary of characteristics that Tables 1 to 3 provide is interesting, but some of those tests match the Table 4 result exactly (Gender for example with a p-value of 0.859 – also note that the reported result in the Abstract about Gender is wrong if this is the correct p-value). It is also important to remember that only when p-values are 1 is there is no evidence of a difference. For example, line 133 has “There was no difference in the arterial pH” – the p-value was 0.421 (the estimated medians and quartiles match exactly which doesn’t make sense with that p-value, so should be checked) which suggests weak evidence of a difference but not “no difference”. Now if the medians do match exactly, then this statement is true but I don’t think that was support for it. Finally, there is the issue of selection bias in the final reported results. If you use p-values to select your final results to discuss, you overstate precision and understate the size of your p-values. Methods for adjusting for this are complex but there is a danger to model selection using p-values and then reporting the p-values as if you knew those were the predictors you were going to look at. Caution should be incorporated in the limitations discussion in both being too definitive in what was identified as being the only predictors that could be important and that what was identified really are the important predictors – this was a very exploratory analysis to consider so many different predictors. Additionally, no interactions were considered, so the assumption of no interactions (or at least no interactions explored) should be noted as that could change results dramatically. I am not saying they have to be, but that is an assumption.

Second, there are four serious issues in the logistic regression models that need to be addressed before this might be publishable as the results hinge on these aspects of the work.

One is that there appears either to be separation issue or something close to it in both some of the univariate logistic regression models and in the reported final model. In Stata, there is supposed to be a check for this (see https://stats.idre.ucla.edu/other/mult-pkg/faq/general/faqwhat-is-complete-or-quasi-complete-separation-in-logisticprobit-regression-and-how-do-we-deal-with-them/ for example). But the estimates are large and the CIs are very wide, which can indicate this issue, at least in some software. Were there any warning messages/automatic modifications when running the Stata logistic regression models? If so, how was that handled? As an example of one of these that I find suspicious for this issue, the final model contained oxygen saturation with an OR CI from -100 to 300. And Figure 2 does not match the numbers reported on line 173 of 1.2 to 105.9 (but the margin labels do, so there is something wrong with the lines displayed in this plot at a minimum).

Another issue in the multivariable logistic regression is that the model building/selection process is unclear. It appears to be a forward selection process of starting with the predictor with the smallest p-value and adding others sequentially. But no details are given on what happens to get the final model from this. I was left to assume that each predictor was tried and then not kept in the model if the p-value was not small after it was added? But after the first predictor was added, you really need to try each one again one at a time to see if its addition is warranted given that new predictor and then pick the one with the smallest p-value each time. This is especially important in the possible presence of multicollinearity of predictors that is likely here (was that checked and what sort of results did it provide?). At a minimum the steps that were taken need to be clarified. Since so much rides on what is in the final model, this process should be carefully documented. With so many predictors, it is challenging to do and report the model building process. The previous suggestion of a multiple testing correction could even mean missing important predictors in this step of the process, so the variable screening before model building might not be the best approach. And with potential separation issues, I would not necessarily recommend starting with all predictors in the model and then dropping them sequentially as important predictors can get large p-values in these situations in some software (although a step-down process could be simpler to implement than step up methods).

An additional assumption in the logistic regression models is that quantitative predictors are linearly related to the response on the logit scale. Was this checked? Not all software makes this possible and if not, then you should note this as an assumption you might make. I could imagine there are some predictors, like pH, that having either too high or too low levels could lead to adverse outcomes. Visualization of the survival response versus the quantitative predictors can help with this (and with the potential for separation issues too) and identify variables that might be candidates for polynomial treatment in the models.

Finally, an issue with all of the logistic regression models is that you have multiple cohorts of subjects being combined. You show this in Figure 1 with the 4 months of subjects and discuss it on lines 188 to 193 which discuss changing standards of care. At least controlling for month as a categorical variable or random effect in all the logistic regression models would help to address this as a potential source of the differences in the survival outcome. With more recent focus on different variants of COVID, is it possible that this played some role in changing survival rates over time (if they were present)? The month variable would likely best not be subject to model refinement but serve as a control variable for considering all the other results.

More minor but important issues:

I am checking the box that data will be available as the authors promise that. Some Plos One authors confuse sharing summary information with sharing the entire data set as used in the models. I am assuming that is the case here. I think this is critical as others work to create meta-analysis of studies like these to synthesize local study results such as this and also because different modeling choices here could lead to very different conclusions.

Abstract should be changed if any of the reported results are modified, which I think likely should happen.

Line 83: Did the patients admitted for other reasons have COVID too? It was unclear from this sentence.

Line 102: Why were the less than 15 year olds dropped? This relates to the limitations on line 257 and should be part of that discussion that these results are limited to subjects over 15.

Line 89: How did you decide between t-test and Mann-Whitney U? Between the parametric Chi-squared test and Fisher’s Exact test?

Table 1: It would great to add some details on each test. A column for the numerical value of the test statistic and another for its distribution (Chi-square (df) or t(df) or “exact”/permutation for the other two versions). This adds quite a bit to the table but can help the reader fully understand what you did. There are some other issues in Table 1: First, the Age^^ suggests it should be mean +- SD but those appear to be medians (Q1 – Q3). Second is how the median results are presented – you are not reporting the IQR, you are reporting Q1 and Q3 with a minus sign between them that is confusing. I would suggest either switching to reporting the IQR = Q3-Q1 or use a comma between them. For Symptoms and Comorbid conditions – are these categories really mutually exclusive? Maybe “Others” means more than one and just wasn’t labelled well – or is it really “Other” things no in the list. I would suggest turning this into a suite of binary variables for each aspect, such as “Fever” or “No Fever” and then each can be explored for relationships to survived/not. Some may not be suitable for use in the tests (like Runny Nose that occurred a total of 5 times or Liver disease that occurred 1 time), but those rare conditions are likely going to cause you issues in the current version with small expected cell counts that violate the assumptions of the parametric Chi-squared test that was used at least for Symptoms. Table 2’s “CXR findings” and Table 3’s “In Hospital complications” also seem like they might not be mutually exclusive categories? After turning these variables into multiple binary variables (possibly), some might be useful in the logistic regression models/persist through the model refinement stages?

Line 136: The directional interpretations (higher/lower) sound like you did one-sided tests. Is that true? If not, then try to report the p-values first and then discuss direction of differences to avoid this confusion.

Line 160: It might be easier to discuss the results if you organize the results by groups – like for categorical predictors separate from quantitative ones? And where possible use the units of the predictors for quantitative variables and have “higher odds” not “high odds”. Maybe even a plot of these results sorted by p-values like in Figure 2 could aid the discussion of them?

Table 4: As noted above, some of these are redundant with previous tests or test a similar null hypothesis (variable vs survived/not) but use a different method. It seems to make some of the earlier work redundant if this is what you really care about – and it seems like this is what you care about most. For oxygen saturation, why was the binary version with a cutoff at 93% used instead the quantitative version of it (both were used before). For the multi-category (analysis of deviance?) tests for things like “Use of Tocilizumab”, some formatting change is needed as the IVIG row looks like part of the previous multi-group test. I am assuming most are from z-tests but some are from Chi-squared tests and all are conditional on all other aspects of the models? Again a little detail on the results can help the reader understand what you did to get those.

Lines 170-175: I think this needs to be expanded. These are your main results. And it is important if your final model contains a suite of predictors to note that these results are conditional/controlled for other variables in the model – as results are different when other variables are not present. And see previous notes on the multivariable logistic regression. Adding visualizations of survived/not versus each of the final model chosen predictors could help the reader understand these results.

Line 265: I think a citation to the software used should be included. Would you be posting your code for the analyses or just the data set? Reproducible research methods that provide code run with output would help readers with other questions about results have access to that although it is not a journal requirement.

General note: I am not sure on PlosOne’s standing on this, but Oxford commas would help in the many lists as some sentences use more complex constructions that get confusing by this absence when it could be used.

Figure 1: The “Referred Out” is the sample analyzed except for the under 15 year olds? How many were in the different months? This would get clarified if month of initial hospitalization were used as a predictor.

There are some minor typographic issues that could be resolved in another round of edits to improve readability and grammatical correctness in a few spots but it is not a poorly written paper.

6. PLOS authors have the option to publish the peer review history of their article (what does this mean?). If published, this will include your full peer review and any attached files.

Reviewer #1: No

Reviewer #2: **Yes: **Christen M Gray

Reviewer #3: **Yes: **Mark C Greenwood

---

## [Author Response · Author response to Decision Letter 0]

31 Mar 2021

We have redone the analysis in the light of reviewer comments- The comments were very comprehensive and detailed hence all responses are given in the Response to reviewers document. It may not be possible to rewrite the details here. Kindly guide us if the journal requires us to do so.

---

## [Decision Letter · Decision Letter 1]

19 Apr 2021

PONE-D-21-00091R1

Determinants of in-hospital mortality in COVID-19; a prospective cohort study from Pakistan

PLOS ONE

Dear Dr. Quratulain Shaikh,

Thank you for submitting your manuscript to PLOS ONE. After careful consideration, we feel that it has merit but does not fully meet PLOS ONE’s publication criteria as it currently stands. Therefore, we invite you to submit a revised version of the manuscript that addresses the points raised during the review process.

ACADEMIC EDITOR: The reviewers have still raised a number of points which we believe major modifications are necessary to improve the manuscript, taking into account the reviewers' remarks. Please consider and address each of the comments raised by the reviewers before resubmitting the manuscript. This letter should not be construed as implying acceptance, as a revised version will be subject to re-review.

We look forward to receiving your revised manuscript.

Kind regards,

Wisit Cheungpasitporn, MD

Academic Editor

PLOS ONE

Reviewers' comments:

Reviewer's Responses to Questions

**Comments to the Author**

1. If the authors have adequately addressed your comments raised in a previous round of review and you feel that this manuscript is now acceptable for publication, you may indicate that here to bypass the “Comments to the Author” section, enter your conflict of interest statement in the “Confidential to Editor” section, and submit your "Accept" recommendation.

Reviewer #1: All comments have been addressed

Reviewer #3: (No Response)

2. Is the manuscript technically sound, and do the data support the conclusions?

Reviewer #1: Yes

Reviewer #3: Yes

3. Has the statistical analysis been performed appropriately and rigorously? 

Reviewer #1: Yes

Reviewer #3: Yes

4. Have the authors made all data underlying the findings in their manuscript fully available?

Reviewer #1: No

Reviewer #3: Yes

5. Is the manuscript presented in an intelligible fashion and written in standard English?

Reviewer #1: Yes

Reviewer #3: Yes

6. Review Comments to the Author

Reviewer #1: l. 28: " A prospective cohort study was conducted [...] to describe the predictors of mortality among hospitalized COVID-19 patients."

"describe" is a really vague objective but using the word predictor suggests that the paper is about P(survival given age).

but at l. 30 "Most non-survivors were above 60 years of age (64%)"

this is about P(Age given survival)

This ambiguity should be resolved.

The method section concerning the multivariate logistic regression could provide more details, in particular regarding the set of candidate explanatory variables.

It is claimed that "all data are fully available without restriction" but the link provided to access the data does not work.

Reviewer #3: Thank you for working to address reviewer comments. It is better but still needs some work to document what was done and why with the variables used and statistical modeling, and the writing is a bit rough in places.

For the response related to Table 4, “Moreover, Linearity assumptions were confirmed for continuous variables and those not fulfilling the assumption were not considered for univariable regression.”

• I did not intend for you to abandon useful predictors because they might show nonlinear relationships. There are options for transformations or polynomials to help with these situations. The papers you reference discuss using plots to check this for at least one way to qualitatively assess for clear missed curvature. This can not confirm no violation of the linearity assumption but can fail to show a problem needs to be addressed.

Potential bias in referrals: “However, the referral was at random depending only on bed availability.”

• This sounds more like haphazard assignment if it is related to openings in beds rather than using a truly random mechanism, like a coin flip or randomization software. It still could have bias because of this. This wording is not used in the paper.

For the revised model selection methods, I like that you have refined this process. I don’t see information on the steps taken (what was in the model to start and what were the removal steps and those p-values – based on the papers (not cited –see below) I suspect they are bigger than 0.25 but the information on the intermediate steps is part of the evidence story) and those should be reported, not just the final model.

There is no clear reason for many of the binary splits in predictor variables used – over 60 years? Over 93% oxygen, Procalcitonin over 2 ng/ml? Were these selected to optimally relate to the response or are the standard cutoffs or were they arbitrarily chosen? It seems like each needs a quick reason for the splits or the variables could have just been used as quantitative predictors (with linearity checks) – is that why they were split because of nonlinearity seen in the relationship to the empirical logits? Or are these standard splits? Or were they are the median or mean of the predictors? This should be explained in the article.

More specific feedback on new version:

You should decide where you are placing references, before or after periods (I think before is best) and be consistent. It is inconsistent throughout the manuscript.

Abstract:

There is a missing a space before “Most”. Extra parenthesis before 21.8. I am not sure what the group is that is being referred to in the vs 7.8% - was that survivors that had 7.8%?

This sentence is in need of at least some punctuation to be more clear: “…, Neutrophil to lymphocyte ratio above 5, procalcitonin above 2ng/ml, unit increase in SOFA score and arterial lactate levels while unit decrease in Pao2/FiO2 ratio and serum albumin were associated with mortality in our patients.”

Data availability:

There is a link posted but it is not active. It looks like this is just waiting until the publication is accepted to become active or the link posted was in error. I appreciate the move to archiving and posting the data set and assume this will get sorted out if the paper is accepted. I am clicking that the data are available under this understanding of the posted inactive link.

Line 69: Change “Till date” to “To this date” or “Up to the time of writing”

Line 71: …, these data or …, these results or …, this information. But data should be a plural word (Plos one may have a policy on this).

Line 90: Data were…

Line 94: I had not run into the term “Purposeful selection…” so you need to cite the two papers here that you are using for the model selection template that are discussed in the response to reviewers. And maybe change “advanced” to “multivariable”. And “likelihood ratio test was” to “likelihood ratio tests were”. You should document the general steps you took to implement this method – it is not a well-enough known process and some of the aspects are arbitrary choices (what p-value cut-off and how the variables are considered for re-entry) and so should be documented for the reader. And here or later on you should note that no interactions among predictors were considered and results might be very different if they were included in the model refinement process.

Line 103: The rest of … and no comma after but and cut “while” and inavailability not in availability?

Line 114: Similar gender distributions were or A similar gender distribution was… in both groups in Table 1

Line 119: First mention of “SOFA score” – no definition or citation for it. It and any other acronyms need to be defined on first use with a citation for its source/definition at a minimum if not explained in the text.

Table 1 and 2: Technically the IQR is the result of calculating Q3-Q1 and a single number. I am assuming you are presenting the (first quartile – third quartile) and the table labels at least need to document that or you can actually report the IQR.

Line 133: capitalize Table

Table 3: There is a ^ symbol but I am not sure what it means as it seems to not be defined?

Table 4: For the Oxygen saturation >93%, a 1 is needed for the Crude OR to be consistent with other categorical predictor presentations. Why is NLR in parentheses and what is it – it is first mentioned on line 143 and not defined. Similarly for CURB 65 and MuLBSTA? Based on the text, it seems like you only tested these predictors versus the response – so you never explored any of the ones? Or you did but are not reporting them? The evidence weakens if you are reporting selected results here. You should document how you chose to report just these predictors.

Line 147: you never defined which predictors were included in the full model – was it all of the Table 4 ones or did you drop gender since it was the only one with a larger p-value reported from the univariate work? Or was it all of the variables discussed in Tables 1, 2, and 3? You should report what what in the initial model and (maybe not required but it would be nice) the order of the variables dropped and possibly re-added in the “purposive” selection process from the papers you mentioned. Also, you note that month was included – but was it used as a quantitative or categorical predictor – and did you see differences based on it too? I am assuming you kept it in all models regardless of its p-value, but making that clear is still needed and a reader might still be interested in changes in survival rates across months. So maybe start with including month in Table 1, 2, or 3 and then in a univariate model in Table 4 and again in Table 5. And if you clarify this early on, you do not need Line 153 with “Final model was adjusted…” – if you keep that, it should be “The final model…” or “All multivariable models were adjusted…”.

Line 154: I don’t usually think of using ROCs to get sensitivity and specificity – that is only one place in ROC at a cutoff of 0.5 for assigning as success/failure. But you can keep this wording if you feel strongly about it.

Table 5: Include the p-values for these terms? The purposive selection should mean all are less than 0.25 but you should still report the p-values that led to their retention in the model.

Line 181: Another acronym not defined (LMIC); Line 187: What is CRS and how is it defined?; Line 198: ARDS?.

Line 200: MODS was defined and that is helpful.

Line 201: Maybe a caveat should be added to these two results to note that they did not end up in the final model. So something like: …, but these predictors were not retained in our final model. And then something to compare what was different about your study/methods than in citation (25) that might have led to different results on these predictors?

Line 207: I know it is the Food and Drug Administration here, but maybe the U.S. FDA?

Line 212: HCQ is Hydroxychloroquine? Not defined again. QTc? And many more than follow – I’ll let the authors carefully check the rest of these.

Line 230: Is this a new paragraph? It is a very long paragraph if not. Maybe think about structure of all these results? Start with reviewing the three predictors retained and then revisit some other intriguing but ultimately abandoned predictors from the univariate results?

Line 259: We plan to add more subjects in the future… or We plan to add more observations in the future…

Line 269: Missing bracket in citation.

Line 279: No details on citation 8 are provided other than year and title.

Please check all other citations carefully as it seems many are missing critical details. For example, I also saw that line 363 had no details on citation 41 other than year and title.

7. PLOS authors have the option to publish the peer review history of their article (what does this mean?). If published, this will include your full peer review and any attached files.

Reviewer #1: **Yes: **Gilles Guillot

Reviewer #3: **Yes: **Mark Greenwood

---

## [Author Response · Author response to Decision Letter 1]

25 Apr 2021

A detailed reply to reviewer's comments is available in the "Response to reviewers" document.

---

## [Decision Letter · Decision Letter 2]

3 May 2021

Determinants of in-hospital mortality in COVID-19; a prospective cohort study from Pakistan

PONE-D-21-00091R2

Dear Dr. Quratulain Shaikh,

We’re pleased to inform you that your manuscript has been judged scientifically suitable for publication and will be formally accepted for publication once it meets all outstanding technical requirements.

Kind regards,

Wisit Cheungpasitporn, MD

Academic Editor

PLOS ONE

Reviewers' comments:

Reviewer's Responses to Questions

**Comments to the Author**

1. If the authors have adequately addressed your comments raised in a previous round of review and you feel that this manuscript is now acceptable for publication, you may indicate that here to bypass the “Comments to the Author” section, enter your conflict of interest statement in the “Confidential to Editor” section, and submit your "Accept" recommendation.

Reviewer #4: All comments have been addressed

Reviewer #5: All comments have been addressed

2. Is the manuscript technically sound, and do the data support the conclusions?

Reviewer #4: Yes

Reviewer #5: Yes

3. Has the statistical analysis been performed appropriately and rigorously? 

Reviewer #4: I Don't Know

Reviewer #5: Yes

4. Have the authors made all data underlying the findings in their manuscript fully available?

Reviewer #4: Yes

Reviewer #5: Yes

5. Is the manuscript presented in an intelligible fashion and written in standard English?

Reviewer #4: Yes

Reviewer #5: Yes

6. Review Comments to the Author

Reviewer #4: It appears that all comments have been appropriately responded to. I have no further comments and recommend publication.

Reviewer #5: This is an interesting study with a huge number of patients and a pleasant outcome. Authors have satisfied the comments of the reviewers

7. PLOS authors have the option to publish the peer review history of their article (what does this mean?). If published, this will include your full peer review and any attached files.

Reviewer #4: **Yes: **Paul W Davis

Reviewer #5: No

---

## [Editor Report · Acceptance letter]

19 May 2021

PONE-D-21-00091R2 

Determinants of in-hospital mortality in COVID-19; a prospective cohort study from Pakistan 

Dear Dr. Shaikh:

I'm pleased to inform you that your manuscript has been deemed suitable for publication in PLOS ONE. Congratulations! Your manuscript is now with our production department. 

Kind regards, 

on behalf of

Dr. Wisit Cheungpasitporn 

Academic Editor

PLOS ONE